# Local and Relayed Effects of Deep Brain Stimulation of the Pedunculopontine Nucleus

**DOI:** 10.3390/brainsci9030064

**Published:** 2019-03-18

**Authors:** Edgar Garcia-Rill, Alan J. Tackett, Stephanie D. Byrum, Renny S. Lan, Samuel G. Mackintosh, James R. Hyde, Veronica Bisagno, Francisco J. Urbano

**Affiliations:** 1Center for Translational Neuroscience, Little Rock, AR 72205, USA; 2Department of Biochemistry and Molecular Biology, University of Arkansas for Medical Sciences, Little Rock, AR 72205, USA; AJTackett@uams.edu (A.J.T.); SBYRUM@uams.edu (S.D.B.); SLan@uams.edu (R.S.L.); MackintoshSamuelG@uams.edu (S.G.M.); 3Department of Biology, Southern Arkansas University, Magnolia, AR 71753, USA; JamesHyde@saumag.edu; 4ININFA, Buenos Aires C1113, Argentina; verobisagno@gmail.com; 5IFIBYNE, CONICET, Universidad de Buenos Aires, Buenos Aires C1113, Argentina; fjurbano@fbmc.fcen.uba.ar

**Keywords:** arousal, CaMKII, cAMP/PK, gamma activity, histone deacetylase, N-type calcium channels, pedunculopontine nucleus, P/Q-type calcium channels, REM sleep, trichostatin A, waking

## Abstract

Our discovery of low-threshold stimulation-induced locomotion in the pedunculopontine nucleus (PPN) led to the clinical use of deep brain stimulation (DBS) for the treatment of disorders such as Parkinson’s disease (PD) that manifest gait and postural disorders. Three additional major discoveries on the properties of PPN neurons have opened new areas of research for the treatment of motor and arousal disorders. The description of (a) electrical coupling, (b) intrinsic gamma oscillations, and (c) gene regulation in the PPN has identified a number of novel therapeutic targets and methods for the treatment of a number of neurological and psychiatric disorders. We first delve into the circuit, cellular, intracellular, and molecular organization of the PPN, and then consider the clinical results to date on PPN DBS. This comprehensive review will provide valuable information to explain the network effects of PPN DBS, point to new directions for treatment, and highlight a number of **issues** related to PPN DBS.

## 1. Introduction

There is sufficient history in the application of deep brain stimulation (DBS) to be confident that implanted electrodes at such sites as the subthalamic region (STN), internal pallidum (iGP), and the pedunculopontine nucleus (PPN) have salutary effects on some of the symptoms of Parkinson’s disease (PD) and related disorders. The problem is that physicians adjust the frequency of stimulation, the duration of pulses applied, and the polarity of the leads activated on an empirical–clinical basis. That is, it appears as if every medical center settles on their individual parameters depending on the response of each patient, without systematic testing across variations in the parameters. Moreover, the symptoms and protocols being studied vary across centers so that blinded testing of beneficial effects are almost never done. Finally, given the large diameter of DBS electrodes, the potential current spread, and the uncertainty in localization, a set of standard criteria that could be applied across the practice needs to be developed. This review provides background information on the anatomy and physiology of the PPN that might explain some of the effects of particular parameters being applied to this region. Such information is essential in formulating a set of effective and replicable parameters that could be applied across patients and hospitals. Very little work on similar morphological and physiological properties of the STN and iGP is available.

## 2. Pedunculopontine Nucleus

### 2.1. Anatomy

One of the first mentions of the reticular formation was in 1887, and was described as “a mass of nerve cells and fibers situated primarily in the brain stem that plays an important role in controlling autonomic functions (such as respiration), reflexive movement, posture and balance, and consciousness and the sleep-wake cycle” [1]. One of the earliest mentions of the pedunculopontine nucleus as part of the reticular activating system (RAS) was in 1909, by Jacobsohn-Lask [2]. As far as links to the basal ganglia, descending projections to the PPN from the globus pallidus were first described using silver degeneration [3]. These projections were confirmed using anterograde transport methods [4]. With the advent of histofluorescence [5] and histochemical [6] methods, the locations of cholinergic neurons were used to delineate the boundaries of the PPN. Descending projections form the pallidum were described as terminating in a region medial to the PPN, the midbrain extrapyramidal area (MEA) [7]. We reported the presence of descending projections from the feline equivalent of the primate iGP, the entopeduncular nucleus, to the locomotion-inducing region around the PPN [8]. We also recorded antidromic responses from the entopeduncular nucleus after stimulation of a locomotion-inducing site, demonstrating a functional (not purely anatomical) link between the iGP and the region of the PPN [8], although these were sparse projections [9], as were ascending projections to the substantia nigra [10]. As far as transmitter-specific cell types, there was an initial disagreement about the presence of PPN cell types other than cholinergic, but triple in situ hybridization studies firmly established that the PPN has cholinergic, glutamatergic, and GABAergic neurons, and there are very few cells with more than one transmitter type [11]. Therefore, while the boundaries of the PPN can be confidently marked by the locations of cholinergic cells, there are two other cell types within the PPN.

For a detailed description of the morphology of these cells, see Reese et al. [12]. This review also lists most of the many projection sites of the PPN, which include thalamic nuclei, mainly in the intralaminar region, as well as substantia nigra, pallidum, lateral hypothalamus, striatum, basal forebrain, and cortex. In addition, the PPN sends descending projections to the pons, medulla, and cerebellum, with very few projections to the spinal cord. The inputs to the PPN originate in the spinal cord and sensory tracts, as well as all of the targets to which it projects [12]. Basically, when the PPN is activated, it will have extensive effects at rostral and caudal regions of the brain. **Issue 1.** As far as the PPN is concerned, its stimulation will have more widespread effects than stimulation of regions that have a more limited number of projection targets than STN or iGP. The question is, what sort of activity do PPN neurons normally manifest?

### 2.2. Physiology

The PPN is part of the RAS, receiving inhibitory inputs from the noradrenergic locus coeruleus (LC) and the serotonergic raphe nucleus (RN). In turn, the PPN excites the LC, but their firing patterns differ across sleep–wake states. The PPN increases firing during waking and REM sleep, and includes bursting activity in the latter state, but the PPN ceases firing during slow wave sleep (SWS). The LC and RN both fire during waking, decrease firing during SWS, and cease firing during REM sleep. This means that the PPN is the only RAS nucleus that fires during waking and REM sleep, two states that manifest high frequency EEG cortical activity [13].

Across the sleep–wake cycle, different PPN neurons manifest different activity. For example, some PPN neurons increase firing rates during rapid eye movement (REM) sleep and are thus referred to as “REM-on” neurons. Some PPN cells fire during both waking and REM sleep and are called “Wake/REM-on” cells, while other PPN neurons fire only during waking and are referred to as “Wake-on” cells [14,15,16,17]. These results demonstrate that PPN neurons are mainly active during high frequency EEG states such as waking and REM sleep. Moreover, injections of glutamate into the rat PPN led to an increased firing during both waking and REM sleep, but injections of NMDA increased firing only during waking, and injections of kainic acid (KA) increased PPN cell firing only during REM sleep [18,19,20,21]. Therefore, NMDA and KA independently activate the states of waking versus REM sleep, respectively. In addition, the intracellular pathways mediating the two states are different, with CaMKII subserving NMDA-induced waking activity and cAMP/PKA subserving KA-induced REM sleep [22,23]. **Issue 2.** Virtually nothing is known about these intracellular pathways and the effects of long-term DBS.

PPN neurons have been reported to fire at beta/gamma frequencies in vivo during waking and REM sleep, but not during SWS [14,15,17,24,25]. A number of labs have performed transection studies showing that separation of the forebrain anterior to the PPN eliminated the manifestation of gamma frequencies in the cortical EEG. However, transections posterior to the PPN showed the presence of cortical EEG gamma activity, and, additionally, PPN stimulation induced high frequencies in the cortical EEG [17,26,27,28,29,30]. Similar results have been reported in vitro, with mouse PPN neurons exhibiting gamma band activity, as well as in a PPN projection target that participates in REM sleep induction [31]. Gamma band activity has been observed in the PPN in the cat in vivo [17], as well as in the region of the PPN in primates [32], and in humans [33] during locomotion. In conclusion, there is considerable evidence from in vivo and in vitro studies, and in various species and in man, for the presence of gamma band activity in the PPN.

PPN neurons were found to fire at ~10 Hz during cortical slow oscillations; in addition, they also support nested gamma oscillations [34]. We demonstrated that PPN cells exhibit a “resting” firing frequency in the 8–12 Hz range in the absence of stimuli, but with the application of depolarizing electrical stimuli, they will fire at gamma frequencies [35,36,37]. However, these higher frequencies are manifested at temperatures close to body temperature, not in the colder environment (~30 °C) typically used for in vitro recordings. Therefore, we perform all of our in vitro recordings at 36–37 °C [36,37,38,39,40,41,42]. The two frequency ranges exhibited by PPN cells, the ~10 Hz resting level and the 20–60 Hz beta/gamma levels were evident in the behaving primate, in which the same PPN neurons fired at resting levels when the animal was quiet, but firing increased to gamma frequencies when the animal awakened or when it walked on a treadmill [32]. **Issue 3.** That is, the same cells fired at gamma band during arousal and locomotor behavior, and at ~10 Hz when in quiet waking, but, such state-dependent activity is not considered in DBS.

### 2.3. Intrinsic Properties

A major discovery in sleep–wake control was the description of the presence of electrically coupled, mainly GABAergic neurons in the RAS [43]. The main clinical implication of this finding is that the atypical stimulant modafinil, used for the treatment of narcolepsy and daytime sleepiness, increases electrical coupling [43,44], which decreases input resistance in GABA neurons and decreases GABA release, thus disinhibiting many regions. This agent is therefore effective in increasing coherence through increased coupling, especially at higher frequencies of activity due to the decrease in GABAergic tone. As such, it represents a potential adjunct to the treatment of neurological and psychiatric disorders that manifest decreases or dysregulation in gamma band activity [45]. **Issue 4.** However, it is not known what effect PPN DBS has on the mechanism of electrical coupling when applied on a long-term basis.

Another major discovery in sleep–wake control was the description of intrinsic gamma band oscillations in PPN neurons, regardless of transmitter type. Briefly, we determined that every PPN neuron fired maximally at 20–60 Hz [35], that all PPN cells possessed intrinsic membrane gamma oscillations [46], and that the mechanism responsible involved high-threshold, voltage-dependent N- and/or P/Q-type calcium channels [36]. Imaging studies revealed that these calcium channels were located all along the dendrites of PPN cells [39], and that a proportion of neurons (~50%) exhibited both N- and P/Q-type calcium channels, another group (~25%) had only N-type channels, and the third group (~25%) had only P/Q-type channels [42,47]. We concluded that these three PPN neuronal types were equivalent to “Wake-REM on” cells since they had both N- and P/Q-type channels, and that cells that “Wake-on” cells were those with only P/Q-type channels, and “REM-on” cells were those with only N-type channels [42,47,48,49]. **Issue 5.** The presence of three physiological types of cells that manifest state-dependent firing patterns complicates the effects of PPN DBS for prolonged periods.

Figure 1 summarizes some of the concepts described above, including the presence of N-type calcium channels that are triggered by kainic acid, participate in REM sleep, and are modulated by the cAMP/PKA intracellular pathway. On the other hand, the P/Q-type calcium channels are triggered by NMDA, participate in waking, and are modulated by the CaMKII intracellular pathway. That is, there are two different calcium channels in charge of two different high frequency activity states, waking versus REM sleep. Moreover, the gamma band activity generated at the level of the cortex manifests coherence in the EEG across distant cortical sites during waking, but not during REM sleep. That is, gamma band activity during waking is different than gamma band activity during REM sleep (see review in [45]).

This helps explain the effectiveness of particular stimulus parameters required to induce locomotion on a treadmill in the decerebrate animal. Shik et al. [50] reported that long duration pulses (0.5–1 ms) delivered at 40–60 Hz could elicit stepping in precollicular-postmamillary transected animals. It should be evident that gamma band activity is a natural resonant frequency for the activity of PPN neurons based on their intrinsic physiological properties. An important additional point in the properties of PPN neurons is that ramp stimuli are essential for depolarizing the membrane, without activating potassium channels that would fight against such depolarization, and are required to activate high-threshold calcium channels. That is why sudden pulses or step stimuli fail to activate high-threshold calcium currents. This also helps explain the need to slowly ramp up current levels when inducing locomotion on a treadmill in the decerebrate animal [49]. **Issue 6.** This provides a window into how sudden versus gradual depolarization of PPN neurons might elicit opposite effects.

### 2.4. Circuit Effects

The ubiquitous presence of gamma band oscillations when PPN neurons are depolarized emphasizes the importance of relaying high frequency activity to ascending and descending targets. One of the main ascending targets of the PPN is the intralaminar thalamus, especially the parafascicular nucleus (Pf) that exhibits electrical coupling of GABAergic neurons [51], as well as P/Q-type calcium channels [36], in its neurons and these are located all along their dendrites [39]. Therefore, gamma frequency activity in the PPN may be relayed to the Pf that in turn generates gamma band activity that is sent to the cortex. This ascending driving by the PPN through the intralaminar thalamus has been proposed to represent bottom-up gamma [52]. Bottom-up processing results from sensory stimuli that activate lower brain centers and the information ascends to higher centers to modulate perception. Top-down processes refer to the imposition by higher centers on the perception and attention to incoming stimuli. It has been suggested that bottom-up and top-down signaling employ different frequency channels, specifically gamma for bottom-up and beta for top-down [53]. **Issue 7.** Assessment of the effects of PPN DBS on perception and attention is needed, since the background of activity provided by DBS on a continuing basis obviously may affect these processes.

What are the effects of generating such high frequency activity in the PPN on its descending targets? Application of electrical or chemical, in particular, cholinergic agonists, to the pontomedullary region will induce decreases in muscle tone at some sites but at other sites elicit locomotor movements [12,53]. Specifically, descending PPN projections to large reticulospinal neurons (presumably involved in the atonia of REM sleep) induce long duration hyperpolarization; however, PPN projections to medium size interneurons induce depolarization that drives spinal locomotor pattern generators [54,55]. High frequency stimulation (>100 Hz) was effective in eliciting this inhibition. Therefore, on the one hand, the PPN inhibits large neurons that drive extensor muscle tone and standing, while, on the other hand, it depolarizes medium-sized neurons that drive locomotion, thus creating push–pull reciprocity between postural drive and locomotor drive [55]. **Issue 8.** The application of continuous PPN DBS may maintain this process, presumably deriving beneficial clinical effects, but this has not been established.

### 2.5. Neuroepigenetics of PPN Activity

Neuroepigenetic processes, such as histone post-translational modifications and DNA methylation, affect the regulation of gene expression in response to a range of environmental stimuli. We determined if intrinsic gamma oscillations in the PPN were modulated by inhibition of histone deacetylation. We showed that acute in vitro exposure to the histone deacetylation inhibitor trichostatin A (TSA) eliminated high-threshold calcium-channel-mediated intrinsic membrane oscillations, particularly in the gamma band range, but not at lower frequencies. Also, pre-incubation with TSA induced a similar decrease in gamma band oscillations, in addition to reducing calcium current amplitudes in PPN neurons. We concluded that there is a specific effect on gamma frequency oscillations if histone deacetylation is blocked, suggesting that gamma oscillations related to arousal normally modulate gene transcription [56]. Figure 2 summarizes our recent findings in which TSA eliminated the manifestation of intrinsic gamma oscillations. TSA is known to block the activity of HDACI in the nucleus and HDACII that can travel into the cytoplasm. In order to narrow down the site of action, we used MS275 to block HDACI and found that it had no effect on gamma oscillations. However, MC1568 specifically blocks HDACIIa and it did block the occurrence of gamma oscillations in PPN neurons. This suggests that the effects of TSA were on HDACII, probably HDACIIa. Normally, then, the manifestation of gamma oscillations through P/Q-type calcium channels that are modulated by CaMKII, presumably affect HDACIIa to affect gene transcription [56].

The proteins modulated normally by gamma band oscillations that were blocked by TSA were related to intracellular calcium regulation, namely, calcineurin (a calcium- and calmodulin-dependent phosphatase), neuronal nitric oxide synthase (enzyme found in PPN cholinergic neurons at high levels), and neuronal calcium sensor protein-1 (over expressed in schizophrenia and bipolar disorder, and modulates gamma oscillations) [41,45]. **Issue 9.** The effects of PPN DBS, or for that matter, any sort of DBS, on gene transcription have never been determined, and would seem of concern when assessing long-term effects.

### 2.6. The Wrong Place for Optogenetics

Recently, certain DBS findings have been questioned by optogenetic experiments, with its users assuming that their findings are correct although they question established results. These investigators fail to self-assess the complexity of their technique and the uncontrolled effects on the preparation being used. Briefly, the optogenetic technique is dependent on inserting, for example, calcium channels into cells that do not normally manifest them, that are opened by light stimulation of opsins. These methods employ microbial (type I) or animal (type II) opsins to target specific populations of cells [57]. While this is a powerful technology, there are a number of caveats that are consistently overlooked. Firstly, the optogenetic technique requires the implantation of an optical fiber to deliver white or blue light. However, blue light has been found to activate cells that do not express the opsins targeted [58]. We do not know how many PPN neurons, or neurons that project to PPN, are activated by such light, mainly because control studies have never been carried out. Secondly, activation of opsins like channelrhodopsin permits studying rapid interactions between cell groups, but their causal role in behavioral events is neither rapid nor direct. This calls into question the interpretation of behavioral assays, for example, measures of sleep–wake rhythms, anxiety, fear, or depression tests, mainly because of the long latency between optic stimulation and the behavioral change. Thirdly, stimulation to induce locomotion, as described above, typically has a latency of ~1 s; however, optogenetic stimulation normally takes 10–20 s to induce an effect [45]. These abnormally long latencies to generate a behavior sheds doubt on the assessment of whether an animal falls sleep or awakens in direct response to optogenetic stimulation. Fourthly, the technique uses the genetic insertion of the opsin using a viral or transgenic vector, which is attached to a promoter that markedly over expresses the channel. That is, the introduction of unknown numbers of channels, e.g., calcium channels, might permanently alter the synaptic activity of these cells as well as the regions to which they project. Typically, “control” studies are carried out to determine if the over expression of these channels will alter simple cell properties such as resting membrane potential. But these studies do not test or report how calcium channel activity, whether low-threshold or high-threshold, might be altered when these calcium “leak” channels are introduced into the cells. Fifthly, the genetic method has changed the targeted brain nuclei by simply altering the lipid environment required for the normal membrane expression of channels and/or receptors [59]. Consequently, recent optogenetic results suggest that the PPN excites, rather than inhibits the startle response, as had previously been established in multiple species and preparations [60]. Another case of recent optogenetic studies questions the established properties the cells play in waking and REM sleep [61].

**Issue 10.** All of these concerns are multiplied given the particular characteristics of PPN cells that make the need to institute tight controls in optogenetic studies especially critical. One example is the presence of high-threshold calcium channels in generating gamma band activity in the PPN, as described above. Activating high-frequency subthreshold oscillations requires the use of a ramp-like, sustained depolarization to reach the activation window of these channels. This effect cannot be achieved using trains of light stimulation. Another factor is localization since normal calcium channels are expressed all along the dendrites of PPN neurons [38,39]. The unmeasured promoter-induced over expression of channelrhodopsin could easily alter the lipid rafts that organize high threshold, voltage-dependent channels [62]. The long-term insertion of these calcium leak channels could gradually change calcium intracellular concentrations, leading to abnormal responses and even cell death in the long run. It is the responsibility of those implementing optogenetic experiments to perform control studies to show that their preparation is not introducing abnormal conditions that create unnecessary controversies.

## 3. Clinical Implications

### 3.1. Stimulus Parameters

In general, DBS involves inserting electrodes in the region of the STN or iGP and delivering high frequency (>80 Hz), short duration (50–100 μs) pulses for as long as 20 of every 24 h. In the stimulation of brain tissue, the rheobase is the lowest current amplitude that can activate the tissue, and the chronaxie is the pulse duration at twice the threshold intensity of the rheobase [63]. The chronaxie of a region with mostly brain fibers is lower than that of a region containing mostly cells [64]. Therefore, stimuli of 50–100 μs in duration preferentially activate fibers, while pulses with durations of 400–1000 μs will preferentially activate neurons. Stimulation frequencies in the range used in STN or iGP DBS would fail to induce firing in cells but instead depolarize neurons beyond firing threshold, essentially blocking their firing (i.e., depolarization block). Thus, the site being stimulated using high frequencies (>100 Hz) was thought to be essentially inactivated, somewhat like a physiological lesion. However, such stimulation may actually modulate neuronal activity orthodromically and antidromically, with little understood consequences at present.

Another factor is electrode size. The stimulating electrodes typically used have a diameter of ~1.2 mm, which in relation to the site being stimulated, represents a sizeable portion of the region. Moreover, it is not known if the insertion of these large diameter electrodes into a region may induce a partial lesion. These conditions require that placement be as accurate as possible, and most centers employ imaging tools to localize the site of stimulation, in addition to determining any beneficial effects during the implantation procedure.

### 3.2. PPN DBS

We carried out a series of studies recording population activity in response to chemical and electrical stimulation of the PPN in brainstem slices. Figure 3 shows power spectra and event-related spectral perturbations (ERSPs) of population recordings in the PPN. Stimulation with the slices with carbachol for 10 min induced gamma band activity as exhibited in the power spectra before (pre) application of TSA. Note the high amplitude peaks at 30–40 Hz. TSA was applied for 10 min, followed by carbachol stimulation for 10 min after TSA (post). Note that the gamma band peaks were all reduced, showing that the overall population activity in the PPN manifested decreases in gamma activation, demonstrating a similar effect at the population level as our previous studies showed at the single-cell level [45].

We then performed a series of studies stimulating brainstem slices containing the PPN using carbachol or low (4 Hz) versus high (40 Hz) frequency stimulation for 30 min. Tandem mass tag-based proteomic analysis of PPN punches in these slices showed sixty proteins changing in abundance by at least two-fold. Eleven of these dysregulated proteins were shared among carbachol, low frequency, and high frequency treatments, which shows shared protein pathway regulation following all treatments. Furthermore, each treatment showed unique proteins dysregulated upon treatment, which indicated that novel protein pathways are dysregulated during each treatment: 8 unique proteins dysregulated for CAR, 10 for low frequency, and 14 for high frequency.

Basically, stimulation at 4 Hz represents activation at EEG delta or sleep-type frequencies, while stimulation at 40 Hz represents activation at the natural frequency of PPN neurons. A major difference between stimulation at these frequencies was the presence of altered levels of actin-cytoskeleton-related proteins after 40 Hz stimulation. This suggests that local high frequency stimulation might have a significant impact on PPN rhythmicity. Indeed, F-actin bundles exert a pushing force on the plasma membrane, stabilizing ion channels membrane expression while interacting with tubulin microtubules [65]. Ca^2+^-calmodulin kinase II (CaMKII) beta and alpha subunit dimerization [66] and their intracellular translocation [67] are dependent on F-actin. Therefore, it would be critical to maintain in the PPN a normal balance between actin and tubulin cytoskeleton protein expression.

In terms of clinical use of PPN DBS, one of the first reports on the use of PPN DBS in PD patients showed that PD patients—who were no longer responsive to l-dopa for their severe axial symptoms and locomotion deficits—responded to PPN DBS with relief of symptoms [68,69]. Further studies used imaging and intracranial recordings to verify implantation sites [70,71,72]. Recordings of population responses in the vicinity of the PPN showed activity in relation to passive and voluntary movements [73], in addition to imaginary locomotor movements [74]. More recent studies showed activity in this region occurred in a phase with alpha oscillations after passive and imagined movements [75].

Another work reported that bilateral stimulation of the PPN decreased falls as well as freezing of gait when using 15 Hz and 25 Hz frequencies [76]. Another group showed improvement in motor scores and falls using 50 Hz and 70 Hz stimulation [77]. One study found that 10 Hz and 25 Hz stimulation showed not only a modest improvement in motor scores but also significant improvement in sleep scores [78,79]. On the contrary, others found that there was no motor improvement but, like prior reports, there were marked improvements in sleep scores and cognitive function [80]. PPN DBS using 20–35 Hz stimulation was found to decrease reaction time, improve falls, and decrease freezing of gait [81]. The same group established that bilateral stimulation was more effective than unilateral stimulation using a double-blind protocol [82]. A comprehensive review of PPN DBS results described the latest surgical and clinical aspects of this field, which includes experience with over 200 patients [83,84]. A very recent MRI-based study determined that the optimal area for PPN DBS was centered on the PPN [85]. These studies used 10–30 Hz stimulation frequencies to alleviate freezing of gait.

While the beneficial effects of PPN DBS to date are encouraging, it must be remembered that they suffer from typically low numbers of patients, there is a considerable variability in postural and locomotor disturbances observed in patients that manifest heterogeneous pathology, and, of course, there exists an uncertainty in the localization of stimulation sites. Adverse events are rare and appear to occur more often after iGP DBS [86]. PPN DBS shows considerable promise, especially since few alternative therapies demonstrate, in addition to salutary effects on gait and postural stability, beneficial effects on sleep patterns and cognitive performance. We suggest that a more thorough assessment of the parameters used in PD patients is needed, particularly in responses to various frequencies of stimulation. As far as PPN DBS is concerned, as discussed above, the natural frequencies of neuronal PPN activity are in the 10 Hz and 20–60 Hz ranges. However, these may represent different functional states that could be used to alleviate different deficits.

Figure 4 shows examples of EEG recordings at various stages of sleep and waking, which are marked by delta frequencies during deep sleep, theta frequencies during light sleep, alpha frequencies in the transition from sleep to waking, and beta and gamma frequencies during full waking. We proposed the presence of a 10 Hz fulcrum marking the border between sleep and waking [87]. As described above, PPN neurons at rest fire at ~10 Hz, but increase firing rates to beta/gamma frequencies when the animal is active and awake. We also know that the RAS exhibits both tonic and phasic activity when modulating postural and locomotor control systems. Therefore, clinical studies need to determine which symptoms and functions respond best to PPN DBS in the 10 Hz frequency range compared to the 20–60 Hz range, and which may be blocked, presumably by depolarization block, by stimulation at high frequencies (>80 Hz). Normally, a standing individual has the knees locked and there is tonic extensor muscle tone. Excessive or insufficient postural tone will result in instability and body sway. However, extensor tone needs to be inhibited in order for leg flexion to begin locomotion. It may be possible to employ PPN DBS at different frequencies to differentially affect standing versus walking, gait versus posture, through differential activation of different descending reticulospinal systems. Testing these ranges of stimulation is critical for optimizing PPN DBS. A novel use of adaptive DBS is to use feedback from brain signals to alter stimulation parameters [88]. A similar approach may help inform the use of different frequencies to guide gait versus posture.

Finally, we should remember that stimulation of the PPN induces activity at a large number of ascending targets. There is reason to conclude that stimulation at gamma frequencies could not only improve stepping but also improve cognition. Given the many proposed roles of gamma activity in the brain, stimulation of the PPN at gamma frequencies could potentiate bottom-up gamma and drive a number of cognitive and perceptual processes. The PPN can be seen as a gamma making machine, and its stimulation could help modulate widespread brain gamma activity. We suggested that PPN DBS could be used not only to normalize waking and REM sleep in PD patients, but also could normalize gamma activity in diseases in which gamma band is dysregulated, including schizophrenia, bipolar disorder, neglect, coma, and even epilepsy [45]. Thorough clinical studies are needed to determine the optimal frequencies of stimulation for the appropriate process being activated or deactivated.

## 4. Conclusions

Delivering stimulation at any site in the brain should take place firm in the knowledge of how the cells at that site will respond to the applied parameters. The PPN is perhaps the only site used for DBS in neurological disorders in which we have detailed information of the properties of its neurons. We know about its electrical coupling and intrinsic membrane properties, and that its natural frequency at rest is 10 Hz, but it is at 20–60 Hz when driving activity. For example, will stimulation at 10 Hz improve postural stability or decrease freezing of gait, and will 40 Hz improve walking? Stimulating the PPN at these natural frequencies are more likely to replicate its normal firing patterns, but not when these are state-dependent. What patterns, changed during the day versus the night, will most improve sleep versus alertness, drowsiness versus attention? Will burst patterns of stimuli, especially at high frequencies, promote REM sleep and the atonia of REM sleep? There is room for improvement in determining if the two ranges of natural frequencies and/or different patterns of stimuli can modulate different motor functions. Moreover, the fact that the PPN projects to so many regions, sending gamma activity to thalamic and forebrain targets thereby generating bottom-up gamma, suggests that DBS might modulate higher functions such as arousal, perception, and attention. A number of issues, therefore, need to be addressed when testing PPN DBS in patients in order to maximize the salutary effects of the therapy.

## Figures and Tables

**Figure 1 brainsci-09-00064-f001:**
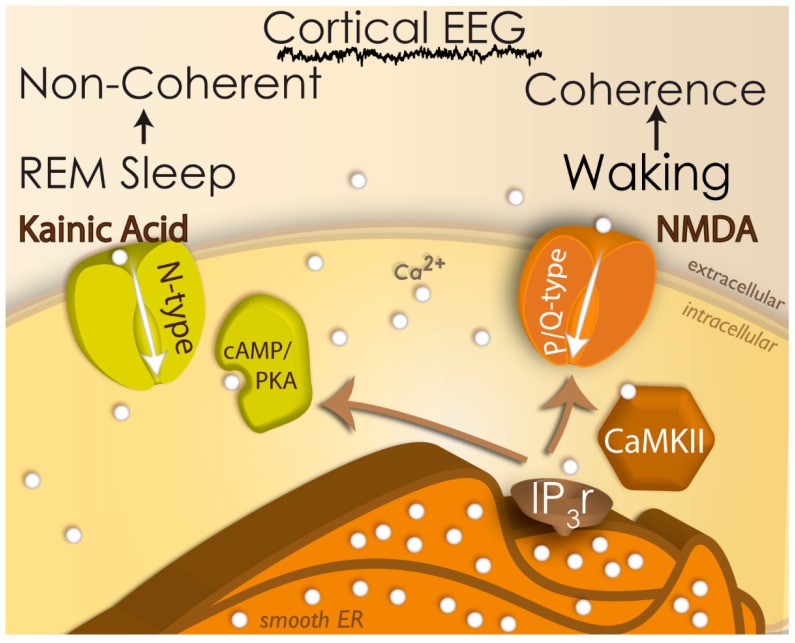
Two states, two channels, two types of gamma EEG. N-type calcium channels (green) are triggered by kainic acid to elicit gamma oscillations during rapid eye movement (REM) sleep, and these channels are modulated by the cAMP/PKA intracellular pathway (green). Gamma oscillations during REM sleep lead to the lack of coherence across distant cortical sites. On the other hand, P/Q-type calcium channels (brown) are triggered by NMDA to induce gamma oscillations during waking, and these channels are modulated by the Ca^2+^-calmodulin kinase II (CaMKII) intracellular pathway (brown). Both intracellular pathways affect calcium release into the cytoplasm by IP3 receptors in the endoplasmic reticulum.

**Figure 2 brainsci-09-00064-f002:**
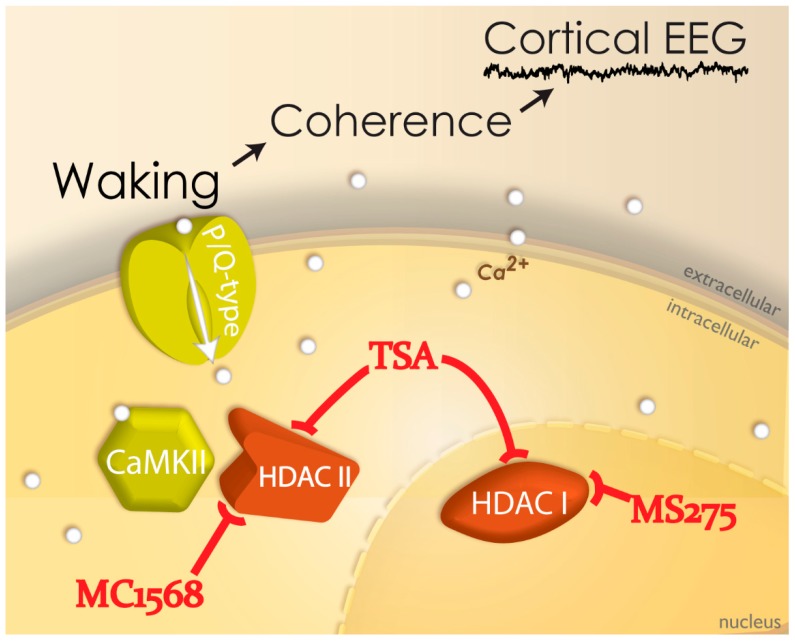
Neuroepigenetic mechanisms behind gamma during waking. Trichostatin A (TSA) was found to block HDACI that remains in the nucleus and HDACII that can travel to the cytoplasm. MS275 blocked HDACI but had no effect on gamma oscillations, while MC1568 blocked HDACIIa and also gamma oscillations. This suggests that TSA blocked HDACIIa in the cytoplasm and generated a deleterious effects on gamma oscillations. Conversely, the manifestation of gamma oscillations through P/Q-type calcium channels during waking, which leads to coherent activity across distant sites in the cortical EEG, modulates gene transcription through HDACIIa.

**Figure 3 brainsci-09-00064-f003:**
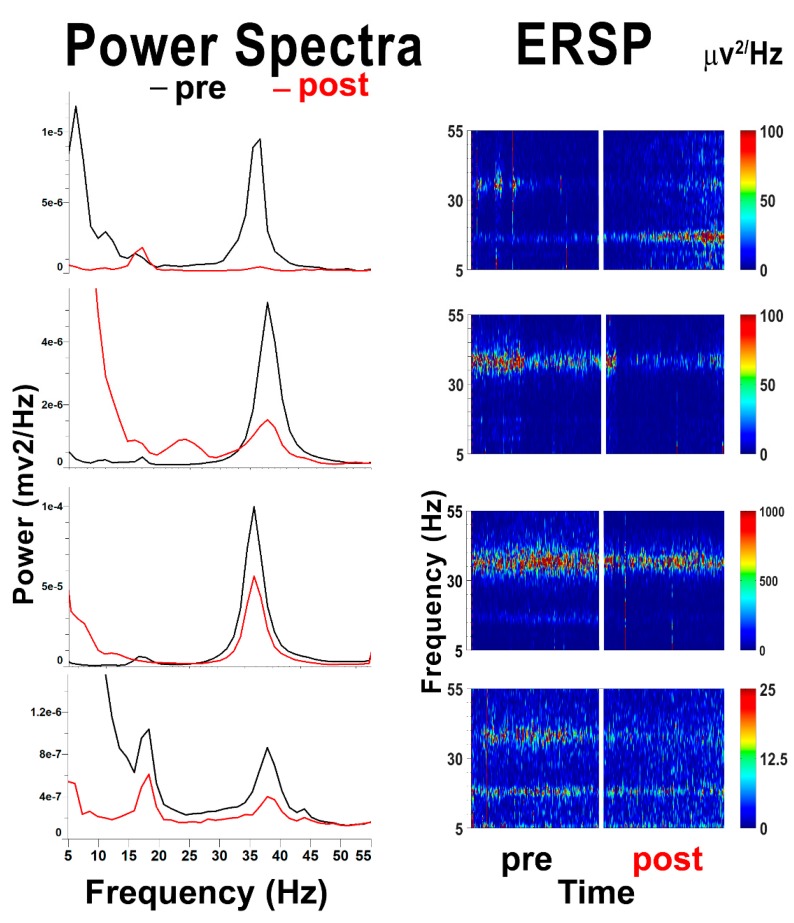
Population activity in the pedunculopontine nucleus (PPN) in response to TSA. The left column shows power spectra of four examples of population recordings from the PPN in brainstem slices. The slices were stimulated with carbachol (50 μM) for 10 min, which induced an increase in gamma band activity in gamma activity (30–40 Hz range) (black lines) before (pre) the addition of TSA. TSA was applied for 10 min and carbachol stimulation continued for 10 min after (post) TSA. The power spectra after TSA (red lines) had reduced gamma activity. These power spectra were taken from the mid-point of carbachol application before and after TSA. The right column shows event-related spectral perturbations (ERSPs), which are basically running power spectra. Note that gamma activity was present in the 30–40 Hz range in each case (left ERSPs), but this activity was reduced after application of TSA for 10 min (not shown, gap in record). Following TSA application, stimulation of the slice with carbachol failed to elicit as large a response. These results suggest that population activity reflects inhibition of gamma activity by TSA similar to what was observed in single cells [45].

**Figure 4 brainsci-09-00064-f004:**
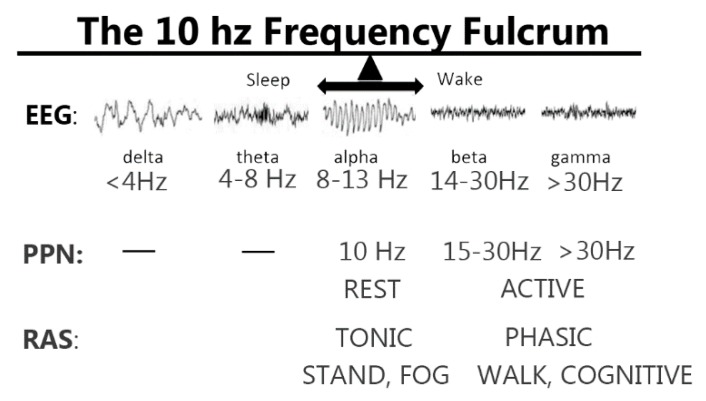
Potential differential effects of PPN deep brain stimulation (DBS). The top row shows examples of EEG recordings during sleep versus waking, with delta (<4 Hz) frequencies occurring during deep slow wave sleep, theta (4–8 Hz) frequencies during light sleep, alpha (8–13 Hz) at the transition between sleep wand waking, beta (14–30 Hz) during waking, and gamma (>30 Hz) during waking. Given the fact that PPN cell activity is ~10 Hz at rest, stimulation at this frequency may have effects related to more tonic activation, including standing and the presence or absence of freezing of gait (FOG). On the other hand, PPN cell activity during active waking is at gamma frequencies, such that stimulation at gamma frequencies may be more related to active processes related to locomotion and arousal.

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
