# Peer review of "Local and Relayed Effects of Deep Brain Stimulation of the Pedunculopontine Nucleus"

_brainsci, 2019, doi:10.3390/brainsci9030064_

Reviewer 1 Report

This paper reviews the effects of deep brain stimulation (DBS) of the pedunculopontine nucleus (PPN).  The review paper outlines the neurobiology of the PPN, including a detailed review of the properties of the neurons in the area, and overviews the major issues with DBS to the PPN.  The paper presents an unbiased assessment of the current state of knowledge of PPN DBS and what is needed to advance the method.  The paper is a significant contribution to the field because it outlines the major mechanism that needs further studied in order to advance the benefits of PPN DBS.

I have only minor comments for the authors:

Line 33: “empirical basis” is ambiguous.  Perhaps say,  “clinical basis” or “empirical-clinical basis;” something along those lines.

Line 106: Citation 17 is missing a bracket.

Line 130:  It reads “this mechanism” — it is not clear which mechanism you are referring to.

Line 285: Should it say “50-100 microseconds?”  It says “ 50-100 sec”.

Line 310: “workers?”  Do you mean researchers or work?

There are fewer than 100 cases published about PPN DBS.  I think that needs to be made clearer. 

Author Response

All of the changes requested were made, including a statement on the fact that over 200 patients have been treated with PPN DBS.  Thank you.

Reviewer 2 Report

Nice and solid science presented here but in a very dense and difficult to follow, almost chaotic way.

The text needs to be better written for one to logically follow and some language editing has to be done, stylistically it reads awkward at times.

The addition of 1 or 2 tables could help.

Moreover, the part referring to DBS falsely mentions that stimulation of the target structure resembles a physiological lesions, a dogma that doe snot hold true any more, as it actually modulates neuronal activity with differential effects on neuraxons and cell somata, as well as orthodromic and antidromic effects. This has to be restated/rephrased.

Author Response

Thank you for the comments, we rephrased the section cited as follows:

   Thus, the site being stimulated using high frequencies (>100 Hz) was thought to be essentially inactivated, somewhat like a physiological lesion.  However, such stimulation may actually modulate neuronal activity orthodromically and antidromically, with at present little understood consequences. 

We are reluctant to add more tables, but believe these changes will promote better understanding by readers.

Reviewer 3 Report

The review article “Local and Relayed Effects of Deep Brain Stimulation of the Pedunculopontine Nucleus” is a summary of the current state of knowledge of the issues of PPN DBS for the treatment of neurological and psychiatric disorders.

Authors provided the description of electrical coupling, intrinsic gamma oscillations, and gene regulation in the PPN in order to suggest the role of PPN in neurological and psychiatric disorders. Authors drafted the review very well, and I do not have any concerns regarding the manuscript.

The following are very minor comments:

Minor comments:

-        Figure 1 legends: It should be green not gren.

Author Response

Thanks for your suggestions. We corrected it.